# A Multi-Level Operation Method for Improving the Resilience of Power Systems under Extreme Weather through Preventive Control and a Virtual Oscillator

**DOI:** 10.3390/s24061812

**Published:** 2024-03-12

**Authors:** Chenghao Li, Di Zhang, Ji Han, Chunsun Tian, Longjie Xie, Chenxia Wang, Zhou Fang, Li Li, Guanyu Zhang

**Affiliations:** 1Electric Power Research Institute of State Grid Henan Electric Power Company, Zhengzhou 450000, China; 2College of New Energy, Harbin Institute of Technology at Weihai, Weihai 264209, China; 3College of Lilac, Harbin Institute of Technology at Weihai, Weihai 264209, China; 2023211583@stu.hit.edu.cn; 4School of Electrical Engineering and Automation, Harbin Institute of Technology, Harbin 150001, China

**Keywords:** virtual oscillator, resilience of power system, extreme weather, preventive control, multi-level operation

## Abstract

This paper proposes a multi-level operation method designed to enhance the resilience of power systems under extreme weather conditions by utilizing preventive control and virtual oscillator (VO) technology. Firstly, a novel model for predicting time intervals between successive failures of the power system during extreme weather is introduced. Based on this, this paper proposes a preventive control method considering the system ramping and transmission constraints prior to failures so as to ensure the normal electricity demand within the system. Further, a VO-based adaptive frequency control strategy is designed to accelerate the regulation speed and eliminate the frequency deviation. Finally, the control performance is comprehensively compared under different experimental conditions. The results verify that the method accurately predicted the time of the line fault occurrence, with a maximum error not exceeding 3 min compared to the actual occurrence; also, the virtual oscillator control (VOC) strategy outperformed traditional droop control in frequency stabilization, achieving stability within 2 s compared to the droop control’s continued fluctuations beyond 20 s. These results highlight VOC’s superior effectiveness in frequency stability and control in power systems.

## 1. Introduction

In recent years, the escalating severity of climate change has led to an increased frequency of extreme weather events, critically impacting the operational stability of power systems [1]. For example, Texas experienced cold weather conditions in 2021, leading to the most significant load shedding in the state’s history. Forty states in the US have reported longer power outages due to destructive storms, and temperatures reaching up to 40 degrees in some southern provinces of China have increased the risk to power systems due to increased electricity demand [2]. In light of the growing challenges posed by extreme weather, the study of power system resilience has become increasingly critical. Resilience refers to the system’s ability to maintain operational functionality and rapidly restore system performance when facing low-probability extreme events [3]. Furthermore, confronted with challenges to power systems’ resilience, it is important to develop effective methods for assessing the risk of failures and implement measures to enhance grid stability and resilience.

Numerous studies have been conducted to enhance the resilience of power systems under extreme weather conditions. Some studies have focused on evaluations of system resilience. These evaluations employ various quantification techniques to identify the strengths and weaknesses of power systems and propose resilience enhancement strategies. For instance, one study introduced a scoring matrix to evaluate system functions from different perspectives [4], while another proposed the resilience “trapezoid” to capture critical system degradation and recovery features across different event phases [5]. Additionally, resilience has been quantified as the difference between the fully functioning system’s capacity and its post-event capacity, offering a more nuanced understanding of resilience impacts [6].

In the realm of operational and planning strategies to enhance resilience, a lot of approaches have been explored. Studies have proposed probabilistic proactive generation redispatch strategies [7] and optimized recovery strategies considering distribution system reconfiguration [8]. A mixed-integer linear programming (MILP)-based method for generation redispatch during ice storms aims to strengthen resilience [9], while a three-stage decision-making methodology evaluates weighted pre-event and post-event performance losses, addressing the spatial uncertainty in fault probabilities [10]. This is complemented by innovative applications of battery energy storage systems (BESSs) for earthquake resilience [11] and the use of machine learning for fault prediction and resilience enhancement, like a distributed offline–online architecture [12] and a multiagent-based hybrid soft actor–critic (HSAC) algorithm [13]. Preventive-action-based strategies, such as multi-sensor prediction-based wide-area monitoring and control [14] and Monte Carlo simulation-based proactive unit commitment frameworks [15], are also gaining prominence in the resilience enhancement discourse.

Furthermore, the integration of renewable energy sources (RESs) has been identified as a crucial factor in augmenting power system resilience. Studies have demonstrated the efficacy of microgrid islanding in creating more resilient distribution systems [16]. A multistage restoration approach that utilizes heuristic optimization algorithms for system reconfiguration and DG islanding has been proposed [17], emphasizing the potential of RESs in emergency response scenarios. This includes approaches like transmission line switching [18], load redispatching [19], and resilience-based unit commitment [20], which collectively underscore the growing importance of RESs in enhancing the resilience of power systems against extreme weather impacts.

The aforementioned methods are crucial for enhancing the resilience of a power system, yet they have limitations. On the one hand, under extreme weather conditions, the power system may experience consecutive failures, potentially leading to insufficient local supply capabilities or even the emergence of localized power supply islands [21,22,23]. In such scenarios, the combined ramping capability of the generators within the local supply area and the power transmission capacity from the external grid may not be sufficient to meet the normal electricity demand within the area. However, current research primarily focuses on emergency control after power system failures, with little consideration given to the impact of potential consecutive failures before they occur. This oversight is particularly evident in the formulation of emergency dispatch strategies for power systems, where there is a lack of comprehensive integration of system ramping and transmission constraints. The system ramping constraint is critical, as it delineates the rate at which a power generator can regulate its output to match fluctuations in electricity demand, given the physical limitation on the rapidity of power output adjustment. Similarly, the transmission constraint encapsulates the capacity limitation of the power transmission network, encompassing a high-voltage transmission line and other grid components, which are pivotal in ensuring the safe and efficient transportation of electrical power without causing overheating or unacceptable voltage drops. It is important that an emergency dispatch strategy accounts for these two constraints to ensure the operational feasibility and resilience of a power system against potential failures.

On the other hand, extreme weather can lead to fragile, isolated local power supply areas with insufficient resources, poor inertia, and low disturbance resistance, threatening frequency stability. Advanced frequency regulation methods are required for their stability. The increasing presence of power electronic interfaces in power grids has enhanced regulation flexibility. However, traditional controls like droop and virtual synchronous machine controls in power electronics, such as energy storage units, are slow to respond to disturbances, particularly in isolated power areas. Recently, virtual oscillator control (VOC) has become important for frequency regulation in power systems rich in power electronics [24,25]. VOC, using the digital control of converters, simulates LC second-order oscillation dynamics. Research on VOC aims to improve inverter performance and practical VOC applications, such as power scheduling in grid-connected modes, islanding support, and enhanced disturbance management. A challenge is balancing VOC’s preference for larger inertia for angular frequency stability against the need for smaller inertia for a better active power response. Thus, a flexible, adjustable VOC beneficial for the frequency response is preferred. Additionally, developing a VOC-based controller that minimizes frequency errors is necessary.

To address the aforementioned issues, this paper proposes a multi-level operation method for improving the resilience of power systems under extreme weather through preventive control and a VO. The new contributions can be summarized as follows:This paper introduces a novel method of preventive control for power systems, particularly focusing on the power constraints between successive failures. This method is significant, as it considers both the combined ramping capability of generators within the local supply area and the power transmission capacity from the external grid before the occurrence of successive failures. This approach is designed to ensure that the normal electricity demand within the local supply area is met, even in the face of potential disruptions. Such a preventive control mechanism is innovative in its comprehensive approach to power system resilience, especially in scenarios of extreme weather and high-risk conditions for cascading failures.The paper proposes a VOC-based adaptive frequency control strategy for power systems. A key element of innovation here is the dynamic adjustment of the VOC loop coefficients according to the system’s operating conditions during transient states. This adaptive strategy marks a significant advancement over traditional VOC implementations, where coefficients are usually static. By dynamically adjusting the VOC loop coefficients, this method enhances the frequency recovery rate, offering a more responsive and efficient approach to frequency regulation under varying system conditions.

The rest of this paper is organized as follows. In Section 2, an upper-level preventive control method that considers successive failures is presented. Section 3 presents a lower-level control method based on a VO to accelerate the frequency regulation speed. A case study is provided in Section 4. Section 5 concludes the paper.

## 2. Upper-Level Preventive Control Considering Successive Failures

Extreme weather can potentially lead to the successive tripping of multiple power lines, and the whole power system might be partitioned into relatively independent regions and even several isolated regions. In this circumstance, some regions may not have an abundant power supply due to the ramping limits of generators and the limited power transmission capacity of lines. This upper-level control framework aims to ensure that the combined capability of the ramping up of generators within the local supply region and the power transmission capacity from the external grid are greater than the load within the local supply region.

During the fault process, the transmission system needs to redispatch generators after the occurrence of faults in order to meet the security conditions of the system under successive failures. The objective function is firstly established according to the dynamic situation of the power system, as shown in (1). The objective function aims to minimize the cost of electricity generation as much as possible while also ensuring minimal variation in the system’s states (voltage and phase angle) before and after any faults.
(1)$$\min C=\sum_{m=1}^{N_{G}}c_{Gm}\left|P_{Gm,t}\right|+\sum_{m=1}^{N_{E}}c_{Em}\left|P_{Em,t}\right|+\left|z_{t}-h(x_{t})\right|$$
where $P_{Gm,t}$ is the active power of the *m*-th generator at time *t*. $c_{Gm}$ is the cost coefficient of the *m*-th generator. $N_{G}$ is the number of generators. $P_{Em,t}$ is the active power of the *m*-th energy storage unit at time *t*. $c_{Em}$ is the cost coefficient of the *m*-th energy storage unit. $N_{E}$ is the number of energy storage units. $z_{t}$ represents the measurement vector including the bus voltage magnitude $V_{i,t}$ ($i\in \Omega_{B}$, and $\Omega_{B}$ is the set of system buses) and phase angle deviation $\theta_{ij}$ ($i\in \Omega_{B}$, $j\in \Omega_{B}$) before the occurrence of the fault. $h(x_{t})$ is the fault state vector including the bus voltage magnitude $V_{i,t}$ ($i\in \Omega_{B}$, and $\Omega_{B}$ is the set of system buses) and phase angle deviation $\theta_{ij}$ ($i\in \Omega_{B}$, $j\in \Omega_{B}$) after the occurrence of the fault.

Furthermore, in the method of preventive control, it is necessary to consider the following constraints:(1)Power flow constraint: The power system must satisfy the power flow equations, and the line flow should be within the prescribed limits [26].
(2)$$\left\{\begin{array}{l} P_{Gi,t}^{\mathrm{all}}-P_{Di,t}^{\mathrm{all}}=\sum_{j\in \Omega_{B}}V_{i,t}V_{j,t}(G_{ij}\cos\theta_{ij,t}+B_{ij}\sin\theta_{ij,t})\\ Q_{Gi,t}^{\mathrm{all}}-Q_{Di,t}^{\mathrm{all}}=\sum_{j\in \Omega_{B}}V_{i,t}V_{j,t}(G_{ij}\sin\theta_{ij,t}-B_{ij}\cos\theta_{ij,t}) \end{array}\right.$$
(3)$$\left\{\begin{array}{l} P_{ij,t}=-{\left(V_{i,t}\right)}^{2}G_{ij}+V_{i,t}V_{j,t}(G_{ij}\cos\theta_{ij,t}+B_{ij}\sin\theta_{ij,t})\\ Q_{ij,t}=-{\left(V_{i,t}\right)}^{2}B_{ij}+V_{i,t}V_{j,t}(G_{ij}\sin\theta_{ij,t}-B_{ij}\cos\theta_{ij,t})\\ P_{ij,\min}\leq P_{ij,t}\leq P_{ij,\max}\\ Q_{ij,\min}\leq Q_{ij,t}\leq Q_{ij,\max} \end{array}\right.$$
where $P_{Gi,t}^{\mathrm{all}}$ and $Q_{Gi,t}^{\mathrm{all}}$ are the active and reactive power injected into bus *i* of all kinds of generation systems, including generators, energy storage units, wind power systems, etc. $P_{Di,t}^{\mathrm{all}}$ and $Q_{Di,t}^{\mathrm{all}}$ are the active and reactive power demands at bus *i* of all kinds of loads. $P_{ij,t}$ and $Q_{ij,t}$ are the active and reactive power from the *i*-th bus to the *j*-th bus. $G_{ij}$ and $B_{ij}$ represent the admittance parameter between transmission line *ij*. $P_{ij,\max}$ and $P_{ij,\min}$ represent the active power limits of transmission line *ij*. $Q_{ij,\max}$ and $Q_{ij,\min}$ denote the reactive power limits of transmission line *ij.*(2)System reserve constraint: The system needs positive and negative spinning reserve capacity to mitigate the impact of uncertainties in wind power and load forecasting, thus ensuring the security and stability of the system operation.(4)$$\left\{\begin{array}{l} \sum_{i=1}^{N_{G}}R_{Gi,t}^{\mathrm{up}}+\sum_{i=1}^{N_{E}}R_{Ei,t}^{\mathrm{up}}\geq \varepsilon_{L}\cdot P_{\mathrm{Load},t}+\varepsilon_{W}\cdot P_{\mathrm{fW},t}\\ \sum_{i=1}^{N_{G}}R_{Gi,t}^{\mathrm{down}}+\sum_{i=1}^{N_{E}}R_{Ei,t}^{\mathrm{down}}\geq \varepsilon_{L}\cdot P_{\mathrm{Load},t}+\varepsilon_{W}\cdot P_{\mathrm{fW},t} \end{array}\right.$$
where $\varepsilon_{L}$ and $\varepsilon_{W}$ represent the proportional forecasting errors of the system load and wind power, respectively. $R_{Gi,t}^{\mathrm{up}}$ and $R_{Gi,t}^{\mathrm{down}}$ denote the positive/negative spinning reserve capacity of generator *i* at time *t*. $R_{Ei,t}^{\mathrm{up}}$ and $R_{Ei,t}^{\mathrm{down}}$ denote the positive/negative spinning reserve capacity of energy storage unit *i* at time *t*. $P_{\mathrm{Load},t}$ denotes the total load within the system, while $P_{\mathrm{fW},t}$ denotes the forecasted wind power output.(3)Constraints on the operation of the generator: These encompass the constraints on the output, the ramping constraints, the startup and shutdown times, and the reserve constraints of generators.
(5)$$P_{Gi,\min}\leq P_{Gi,t}\leq P_{Gi,\max}$$
(6)$$\left\{\begin{array}{l} T_{Gi,t}^{\mathrm{on}}\geq T_{Gi,\min}^{\mathrm{on}}\\ T_{Gi,t}^{\mathrm{off}}\geq T_{Gi,\min}^{\mathrm{off}}\\ P_{Gi,t+1}-P_{Gi,t}\leq v_{\mathrm{Ramp},Gi}^{\mathrm{up}}\cdot \Delta t\\ P_{Gi,t}-P_{Gi,t+1}\leq v_{\mathrm{Ramp},Gi}^{\mathrm{down}}\cdot \Delta t \end{array}\right.$$
(7)$$\left\{\begin{array}{l} P_{Gi,t}+R_{Gi,t}^{\mathrm{up}}\leq P_{Gi,\max}\\ 0\leq R_{Gi,t}^{\mathrm{up}}\leq v_{\mathrm{Ramp},Gi}^{\mathrm{up}}\cdot \Delta t\\ P_{Gi,t}-R_{Gi,t}^{\mathrm{down}}\geq P_{Gi,\min}\\ 0\leq R_{Gi,t}^{\mathrm{down}}\leq v_{\mathrm{Ramp},Gi}^{\mathrm{down}}\cdot \Delta t \end{array}\right.$$
where $P_{Gi,\max}$ and $P_{Gi,\min}$, respectively, denote the upper and lower limits of the active power output of generator *i*. $T_{Gi,t}^{\mathrm{on}}$ represents the startup time of generator *i* during time period *t*, while $T_{Gi,t}^{\mathrm{off}}$ represents the shutdown time of generator *i* during time period *t*. $T_{Gi,\min}^{\mathrm{on}}$ denotes the minimum allowable startup time for generator *i*, and $T_{Gi,t}^{\mathrm{off}}$ represents the minimum allowable shutdown time for generator *i*. $v_{\mathrm{Ramp},Gi}^{\mathrm{up}}$ and $v_{\mathrm{Ramp},Gi}^{\mathrm{down}}$ represent the upward and downward ramp rates of generator *i*, respectively. Δt denotes the minimum scheduling period.(4)Constraints on the operation of the energy storage unit: These encompass constraints on the reserve and capacity of energy storage units.
(8)$$\left\{\begin{array}{l} 0\leq R_{Ei,t}^{\mathrm{up}}\leq P_{N,Ei}-P_{Ei,t}\\ 0\leq R_{Ei,t}^{\mathrm{down}}\leq P_{Ei,t}+P_{N,Ei} \end{array}\right.$$
(9)$$\left\{\begin{array}{l} E_{\min,i}\leq E_{t,i}-(P_{Ei,t}+R_{Ei,t}^{\mathrm{up}})/\eta_{i}\leq E_{\max,i}\\ E_{\min,i}\leq E_{t,i}-(P_{Ei,t}-R_{Ei,t}^{\mathrm{down}})/\eta_{i}\leq E_{\max,i} \end{array}\right.$$
where $P_{N,Ei}$ represents the rated power of energy storage unit *i*, while $E_{\min,i}$ and $E_{\max,i}$ represent the minimum and maximum energy capacities of storage unit *i*. $\eta_{i}$ represents the charge or discharge efficiency of storage unit *i*.(5)Power constraint between successive failures: Following a power system failure, isolated operation in certain regions or limited interconnection with the rest of the large power grid may ensue. In such scenarios, the affected area may experience insufficient generation capacity. In particular, in the intervals between consecutive failures, without a well-orchestrated generation plan, generators within the region may struggle to swiftly adjust their output to anticipate the next contingency. Additionally, due to transmission constraints, the remainder of the large power grid may be unable to supply adequate power to this area, thereby increasing the risk of widespread power outages in the power system.

We utilize Figure 1 for a more intuitive explanation. Assuming that Node 9 bears a substantial load at time $t_{0}$ while Node 3 exhibits a limited generation level at the same time, at this juncture, the load at Node 9 will be sustained by Lines 3–9, 6–9, and 8–9. Furthermore, employing the approach outlined in Section 2, if it is possible to forecast the occurrence of faults on Lines 6–9 and 8–9, and the time of occurrence for each fault is denoted by $t_{1}$ and $t_{1}+K\Delta t$, respectively, it becomes evident that, following the fault on Line 6–9, the load at Node 9 will be supported by Lines 3–9 and 8–9. Subsequently, after the fault on Line 8–9, the load at Node 9 can only be supplied by Line 3–9. If the ramping capability of the generator at Node 3 is insufficient to meet the load requirement at Node 9 at time $t_{1}+K\Delta t$, it becomes necessary for Node 9 to curtail a portion of its load to maintain supply–demand equilibrium. One potential resolution is to preemptively increase the output of the generator at Node 3 upon forecasting the fault occurrences on Lines 6–9 and 8–9 at times $t_{1}$ and $t_{1}+K\Delta t$, thereby mitigating Node 9’s reliance on Lines 6–9 and 8–9 for supply and ensuring that the generator’s output at Node 3 is able to ramp up to meet the load requirements at times $t_{1}$ and $t_{1}+K\Delta t$.

Based on the aforementioned analysis, the power constraint between successive failures is set as follows:(10)$$\left\{\begin{array}{l} \sum_{i\in \Omega_{\mathrm{ISO}}}\left(P_{Gi,t}+R_{Gi,t}^{\mathrm{up}}\right)+\sum_{ij\in \Omega_{\mathrm{CON}}}P_{ij,\max}\geq \sum_{i\in \Omega_{\mathrm{ISO}}}P_{Di,t}^{\mathrm{all}}\\ \sum_{i\in \Omega_{\mathrm{ISO}}}\left(P_{Gi,t+1}+R_{Gi,t+1}^{\mathrm{up}}\right)+\sum_{ij\in \Omega_{\mathrm{CON}}}P_{ij,\max}\geq \sum_{i\in \Omega_{\mathrm{ISO}}}P_{Di,t+1}^{\mathrm{all}}\\ \ldots \\ \sum_{i\in \Omega_{\mathrm{ISO}}}\left(P_{Gi,t+K}+R_{Gi,t+K}^{\mathrm{up}}\right)+\sum_{ij\in \Omega_{\mathrm{CON}}}P_{ij,\max}\geq \sum_{i\in \Omega_{\mathrm{ISO}}}P_{Di,t+K}^{\mathrm{all}} \end{array}\right.$$
where K represents the time interval between adjacent failures. $\Omega_{\mathrm{ISO}}$ denotes the set of nodes in the local power supply area, while $\Omega_{\mathrm{CON}}$ signifies the set of lines connected to the local power supply area.

The equation illustrates that, following a failure, the power system may be partitioned into relatively independent regions. It is imperative to ensure that, during the occurrence of a failure, the load within the local power supply area remains unaffected. Specifically, the combined ramping capability of the generators within the local supply area and the power transmission capacity from the external grid must be sufficient to meet the normal electricity demand within the area.

(6)Constraint on renewable energy output: The actual dispatch output of wind power must not exceed the forecasted wind power output.
(11)$$0\leq P_{\mathrm{sW},t}\leq P_{\mathrm{fW},t}$$
where $P_{\mathrm{sW},t}$ represents the actual output of wind power.

To sum up, the power flow and generator/energy storage system operation constraints are critical for ensuring that the physical operation of the power system adheres to technical specifications and limits. The constraints of the system reserve and power stability between failures are vital for preparing the system against uncertainties (like wind power and load forecasting errors) and for maintaining stability and continuous operation in the aftermath of system failures, ensuring that there is sufficient generation capacity and strategic operation planning to avoid widespread outages. The constraint on renewable energy output focuses on aligning the actual renewable power dispatch with forecasted outputs. By combining the above six constraints and the objective function, the upper-level preventive control scheme considering successive failures can be obtained.

Notably, in the preventive control optimization process, we utilize a differential evolution–particle swarm optimization (DE-PSO) algorithm [27]. The DE-PSO algorithm, merging differential evolution (DE) and particle swarm optimization (PSO), excels in optimizing complex problems without easily getting trapped in local optima. Its hybrid nature combines DE’s global search with PSO’s swift convergence, enabling the effective exploration and exploitation of the search space. This makes DE-PSO particularly adept at handling problems with numerous constraints, as it can adaptively balance between exploring new areas and refining existing solutions. Thus, DE-PSO stands out for its ability to navigate challenging optimization landscapes and find high-quality solutions amidst a multitude of constraints. Considering the dynamic and often rapidly changing conditions of power systems during extreme weather events, the optimization is set to recur at intervals of 15 min. This interval is optimal for updating the system’s preventive strategies in response to evolving environmental and operational conditions, thereby enhancing the system’s adaptability and resilience against cascading failures. Testing indicates that the PSO algorithm can complete an optimization cycle in approximately 10 s, and this duration is acceptable and ensures a balance between computational expediency and precision.

## 3. Lower-Level Control Based on Virtual Oscillator

On short time scales, the stochastic nature of the load and renewable energy sources may influence the stability of the system frequency, especially when the power system is divided into relatively independent regions due to faults. This section (Section 3) explores the rapid adjustment characteristics of power electronic devices of energy storage devices and proposes a frequency control method for power systems based on a VO.

Firstly, the integration of VOC into a multi-level frequency control architecture marks a significant advancement for power systems, particularly those rich in power electronics. This approach represents a pioneering step in harmonizing VOC strategies with contemporary power system control requirements. Secondly, we developed an innovative frequency regulation method using the VOC strategy. This method utilizes a rapid time-domain model in conjunction with coupling droop characteristics, enabling VOC-controlled inverters to measure frequency without the need for a Phase-Locked Loop (PLL). This advancement significantly reduces the computational load, enhancing the efficiency of the system. Thirdly, this manuscript details a novel methodology for optimizing VOC parameters, including virtual capacitance and inductance. This optimization is crucial for dynamically adjusting system parameters in response to real-time operational conditions, thereby improving the frequency regulation performance of power systems [28].

### 3.1. Characteristic Analysis of VOC Inverter

Figure 2 depicts a typical VOC inverter of the Van der Pol (VdP) type, comprising a VdP type of VO, DG, and an inverter. The circuit model of the VdP type of VO consists of three parallel parts: (1) an LC resonance loop that can generate a resonant frequency and whose oscillation characteristics are reflected in the bidirectional energy flow between a virtual capacitor and a virtual inductor; (2) a negative impedance, which is essential to the system’s transient process and closely related to the inertial frequency regulation scale; (3) a voltage-controlled current source.

The dynamics of VOC are given by [29]
(12)$$\left\{\begin{array}{l} \frac{di_{L}}{dt}=\frac{v_{c}}{L}\\ \frac{dv_{c}}{dt}=-\frac{\alpha }{C}v_{c}^{3}+\frac{\sigma }{C}v_{c}-\frac{1}{C}i_{L}-\frac{\kappa_{i}}{C}i \end{array}\right.$$
where $i_{L}$ is the inductor current. $v_{c}$ is the capacitor voltage. C, L, and σ are the virtual capacitance, inductance, and conductance, and the nonlinear voltage-dependent current source is parameterized by α. The controller is interfaced via voltage and current scaling values, $\kappa_{v}$ and $\kappa_{i}$.

From Figure 2, the output voltage command is
(13)$$v=\kappa_{v}\left(v_{C}\cos\Phi -\varepsilon i_{L}\sin\Phi \right)$$
where $\varepsilon =\sqrt{L/C}$, and Φ is a fixed user-defined angle, typically set to either 0 or π/2, which specifies the type of droop-like characteristic that the VO-controlled inverter exhibits in the steady state.

Further, assuming Φ=π/2, an averaged model is utilized by averaging the trajectory of the oscillator over one ac cycle, and then we can obtain the following amplitude and frequency dynamics:(14)$$\frac{d{\bar{\theta }}^{*}}{\;dt}=-\frac{\kappa_{v}\kappa_{i}}{6C{\bar{V}}^{2}}P$$
where P is the average three-phase real power over one cycle $\omega^{*}/2\pi$.

Given (14) and the fact that $\theta \left(t\right)+\omega t=\theta^{*}\left(t\right)+\omega^{*}t$ and dθ/dt=0 in the steady state, we have
(15)$$\omega_{\mathrm{eq}}=\omega^{*}-\frac{\kappa_{v}\kappa_{i}}{2C{\bar{V}}_{\mathrm{oc}}^{2}}P_{\mathrm{eq}}$$

This VOC inverter uses symmetrical pulse-width modulation (SPWM) in the control. A reference signal (usually a sine-wave signal) is compared with a high-frequency triangular-wave signal, and the on-time of the switching device is controlled according to the comparison results.

Assuming that the reference signal is $V_{\mathrm{ref}}$ and the high-frequency triangular-wave signal is $V_{\mathrm{tri}}$, their comparison yields a result. If $V_{\mathrm{ref}}$ > $V_{\mathrm{tri}}$, the switching device conducts; if $V_{\mathrm{ref}}$ < $V_{\mathrm{tri}}$, the switching device turns off.

By controlling the conduction time of the switching devices, the output voltage can be regulated. The longer the conduction time, the higher the output voltage; the shorter the conduction time, the lower the output voltage. Specifically, an SPWM signal can be derived from the comparison of $V_{\mathrm{ref}}$ and $V_{\mathrm{tri}}$ to control the conduction time of the switching devices. This modulation signal can be described by the following formula:(16)$$D=\frac{V_{\mathrm{ref}}}{V_{\mathrm{tri}}}\times 100\%$$
where *D* is the duty cycle representing the percentage of conduction time in the total period, $V_{\mathrm{ref}}$ represents the reference signal, and $V_{\mathrm{tri}}$ represents the triangular-wave signal. By controlling this duty cycle, the precise adjustment of the output voltage can be achieved, thus realizing SPWM control.

The converters based on VOC exhibit deviations from their nominal responses, which vary depending on the active power. These deviations are characterized by the slope of the droop curves, indicating that VOC droop characteristics are embedded in its nonlinear dynamics model. As a result, VOC has the primary frequency regulation characteristics of droop control. The VOC time-domain model is rapid, and when combined with its coupling droop characteristics, it enables the measurement of frequency without requiring the use of a Phase-Locked Loop (PLL). This feature significantly reduces the calculation burden of the controller. Additionally, the primary frequency regulation scale of VOC is shorter than that of droop control, allowing for faster attainment of the steady state.

### 3.2. Control Method

#### 3.2.1. Overall Framework

This paper proposes a multi-level frequency control architecture suitable for a VOC strategy in power systems, as depicted in Figure 3. The proposed architecture consists of two levels: lower-level primary frequency control and upper-level secondary frequency control. At the lower level, a multistage frequency regulation architecture is used with VOC to achieve frequency regulation. Additionally, a virtual link is added to compensate for lower inertia power, and the dynamic adjustment of the virtual capacitance is performed to optimize the inertia of the frequency regulation dimension. To facilitate information interaction between distributed generation units, network layer communication is utilized.

#### 3.2.2. Determination of Virtual Capacitance

Figure 4 shows the transient frequency characteristics of the power system before and after optimal inertia frequency regulation optimization. The red curve is the frequency characteristics before optimization, and the blue curve is the frequency characteristics after optimization. Time scales $t_{1}$ and $t_{3}$ are the frequency dip and frequency recovery scales before optimization, while time scales $t_{2}$ and $t_{4}$ are the frequency dip and frequency recovery scales after optimization, respectively. At the scale of $t_{1}$ or $t_{2}$, the frequency change rate rises sharply after the disturbance appears and gradually decreases within this time scale. However, at the $t_{3}$ or $t_{4}$ scale, the frequency change rate gradually increases. This paper expects a longer frequency drop process with a smaller frequency deviation, that is, $t_{2}>t_{1}$ and $\Delta f_{2}<\Delta f_{1}$. Also, the frequency recovery process is expected to be faster, that is, $t_{4}<t_{3}$.

Based on the analysis of the influence of VOC parameters on the control transient characteristics in Appendix B, it was found that virtual capacitance plays a crucial role and has a proportional relationship with the transient response time. In order to achieve the ideal optimization effect, the value of virtual capacitance should be larger during the frequency drop stage but smaller during the frequency recovery stage. To meet the requirements for inertia frequency regulation optimization, an adaptive virtual inertia control strategy based on the dynamic change in VOC parameters is proposed in this article, which is
(17)$$\left\{\begin{array}{l} C_{inertia}=k_{J}\frac{\left|\omega -\omega_{0}\right|}{\omega -\omega_{0}}\frac{d\Delta \omega }{dt}\\ C_{con}=C_{0}+C_{inertia}\\ L=\frac{1}{C{\left(\omega_{0}\right)}^{2}} \end{array}\right.$$
where $C_{inertia}$ is the value of the virtual compensation capacitance. $C_{0}$ is the initial value of the virtual capacitor. $C_{con}$ is the final virtual capacitance value. Δω is the frequency variation. $\omega_{0}$ is the rated angular frequency. $k_{J}$ is the inertia compensation coefficient.

By substituting (16) into Equation (15), the following equations can be obtained:(18)$$\frac{2\left(C_{0}+C_{inertia}\right){\bar{V}}_{\mathrm{oc}}^{2}}{k_{v}k_{i}}\left(\omega -\omega *\right)=P_{\mathrm{out}}$$
(19)$$\frac{2C_{inertia}{\bar{V}}_{\mathrm{oc}}^{2}}{k_{v}k_{i}}\left(\omega -\omega *\right)=P_{\mathrm{out}}-\frac{2C_{0}{\bar{V}}_{\mathrm{oc}}^{2}}{k_{v}k_{i}}\left(\omega -\omega *\right)$$

Equation (17) represents the active power frequency sag equation of VOC, with the right side showing the coupling effect between active power and frequency, while the left side represents the superimposed virtual inertia power. The adaptive control method of virtual inertia allows for the automatic adjustment of the virtual capacitance based on changes in frequency variation. When the frequency change rate is high, a higher value of virtual capacitance can be set to increase the transient response time and reduce the frequency change rate simultaneously. As the frequency approaches stability, the frequency change rate gradually decreases, and the value of the virtual capacitance decreases accordingly, thus accelerating the transient response speed of the system.

#### 3.2.3. Determination of Virtual Inductor

In Appendix B, we analyze the coupling droop characteristics of the VOC inverter, and here, we assume that the coupling droop relation of active power and frequency is
(20)$$\left\{\begin{array}{l} \omega_{i}=\omega *-m_{P-\omega }{\bar{P}}_{i}\\ m_{P-\omega }=\frac{k_{v}k_{i}}{2C_{con}{\bar{V}}_{\mathrm{oc}}^{2}} \end{array}\right.$$
where $m_{P-\omega }$ is the droop coefficient. $\omega_{i}$ is the angular frequency for node *i*. ${\bar{P}}_{i}$ is the active power of node *i*. ω* is the rated angular frequency.

As shown in (19), the droop coefficient of VOC is determined by the VOC parameters. Therefore, dynamically adjusting the VOC parameters can be used to adjust the frequency characteristics of a power system. Based on sparse communication networks, the upper-level control should be incorporated into dynamically adjustable VOC parameters. Assuming that the power-generating unit acts as a node, the adjacency matrix A forms a communication network connection between nodes *i* and *j*, with its elements $a_{ij}=a_{ji}>0$ describing the network’s communication status. The upper-level frequency control based on VOC and consensus theory is as follows:(21)$$\omega_{i}=\omega *+\frac{k_{v}k_{i}}{2C_{con}{\bar{V}}_{\mathrm{oc}}^{2}}{\bar{P}}_{i}-\Omega_{i}$$
(22)$$\frac{d\Omega_{i}}{dt}=-g_{f}\left(\omega_{i}-\omega *\right)-\overset{n}{\underset{j=1}{\sum }}a_{ij}\left(\Omega_{i}-\Omega_{j}\right)$$
where $\Omega_{i}$ is the control variable of the frequency controller. $g_{f}$ is the positive gain that affects the frequency adjustment speed of node *i*. $a_{ij}$ is the positive gain that affects the power-sharing rate between nodes *i* and *j*.

According to consensus theory, the state variables of both $\Omega_{i}$ and $\Omega_{j}$ will eventually converge to the same value. Specifically, each node of the droop characteristic curve will move by the same amount, $\omega_{i}-\omega_{0}$, as depicted in Figure 5. Therefore, there is an agreement in the change in droop characteristics across all nodes.

When the system reaches a steady state, the derivative on the left-hand side of (21) becomes zero. This is also the point where the frequency deviation in (21) equals zero, which means that the system frequency and the rated angular frequency are equal to each other. Therefore, the system regulation achieves no difference, i.e., $\omega_{i}=\omega *$. To control the upper-level controller, the VOC parameter is introduced, and the virtual capacitor *C* remains unchanged. On the other hand, the virtual inductor *L* can be adjusted, which results in the system frequency being determined by the adjustable virtual inductor $L_{adj}$. Based on the adjustable virtual inductor $L_{adj}$, the system frequency is
(23)$$\omega_{i}=\frac{1}{\sqrt{L_{adj}C_{con}}}$$

Assuming that the parameters of the LC resonance loop are $L_{con}$ and $C_{con}$ when the system frequency is the rated value, by substituting (22) into (20), one can obtain
(24)$$L_{adj}=1/C_{con}{\left(\frac{1}{\sqrt{L_{con}C_{con}}}+\frac{k_{v}k_{i}}{2C{\bar{V}}_{i}^{2}}{\bar{P}}_{i}-\Omega_{i}\right)}^{2}$$

As can be seen from (23), the control variable $\Omega_{i}$ of the controller is absorbed into the coupling droop characteristics of VOC through the adjustable inductance of the VO controller. When the system reaches a steady state, the adjustable inductance $L_{adj}$ value will tend to be $L_{con}$, and the frequency $\omega_{i}$ will also tend to be the rated frequency ω*.

## 4. Case Study

### 4.1. System Description

This paper utilized the IEEE-39 power system in the simulation experiments. This system comprises 38 transmission lines, 10 generators, 3 wind farms, and 6 energy storage units. A detailed visualization of the power system is presented in Figure 6. The parameters of the generators are listed in Table 1. It is assumed that the wind speed in the wind farm follows a Weibull distribution, and the scale factor, shape factor, cut-in wind speed, rated wind speed, and cut-out wind speed are listed in Table 2. The parameters of the energy storage units are listed in Table 3. VOC is employed in the inverters of the energy storage units, and they are coordinated through the cyber network layer shown in Figure 7. The parameter settings of the VO controllers for the energy storage units are listed in Table 4.

### 4.2. Verification of Upper-Level Preventive Control

Suppose that high wind occurs at 9:00 throughout the power system, causing Line 14–15, Line 9–39, and Line 3–4 trips in succession. This results in the isolated operation of the local power grid, as shown in the following Figure 8. The occurrence process of successive failures that is measured in the actual power system is as follows:(1)At 9:28, Line 14–15 is broken and fails to reclose;(2)At 10:02, Line 9–39 is broken and fails to reclose;(3)At 10:42, Line 3–4 is broken and fails to reclose, and the whole power system is separated into two regions, as shown in Figure 8.

**Figure 8 sensors-24-01812-f008:**
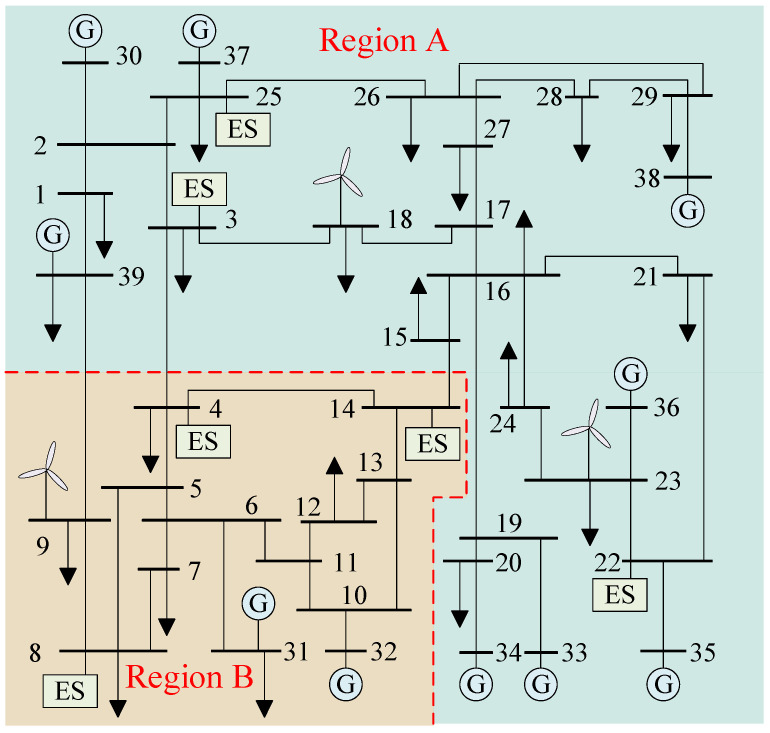
Separated power system after successive failures.

To verify the effectiveness of the model of successive failures in the power system, this whole successive failure process is modeled by the proposed method according to Equations (A2) and (A3).

In Equations (A2) and (A3), the expectation value of the Poisson distribution λ is a key parameter. It represents the average rate of occurrences (such as line outages) per unit time. To determine λ precisely, historical data on the frequency of line outages, particularly under specific weather conditions, are required. These data form the basis for a reliable estimate of λ. The calculation of λ involves the following two steps:
(1)Data Collection: Gather historical data on line outages. These data should include the number of outages that occurred over a significant period, along with the duration of each observation period. It is essential to consider the specific weather conditions during which these outages occurred, as λ may vary with different environmental factors.(2)Calculating the Average Rate: Divide the total number of observed outages by the total observation time to calculate the average rate of outages per unit time. The formula for this calculation is as follows:(25)$$\lambda =\frac{Total\;number\;of\;outages}{Total\;observation\;time}$$

After the determination of λ, the probability distribution and probability density function of the lines can be calculated. As a result, Figure 9 and Figure 10 show the probability distribution and probability density function of Line 3–4, Line 9–39, and Line 14–15. Furthermore, the time interval between successive failures is obtained, as shown in Figure 11. Given a confidence level of 1−α=0.95, the failure time of Line 14–15 is calculated as 9:30, the time interval $t_{L,1-\alpha }$ between the successive failures of Line 14–15 and Line 9–39 is calculated as 30 min according to Equation (A7), and the time interval $t_{L,1-\alpha }$ between the successive failures of Line 9–39 and Line 3–4 is calculated as 45 min according to Equation (A7). After the successive failures of Line 14–15, Line 9–39, and Line 3–4, the whole power system is separated into two regions. It is evident that the proposed method accurately predicts the time of the line fault occurrence, with a maximum error not exceeding 3 min in comparison to the actual occurrence. Therefore, the proposed method is capable of providing a relatively precise estimation of the occurrence time of transmission line faults in the power system, thus furnishing essential prospective fault information for subsequent preventive control measures.

Furthermore, based on the predicted fault occurrence time of the line, the method described in Section 2 is applied for the preventive control of the power system, with a time interval of 15 min between adjacent schedules. Figure 12 depicts the output of generators and energy storage units from 9:00 to 10:45, while Figure 13 illustrates the active power flows in the transmission lines connecting Region A and Region B. It is observed that at 9:00, the flow through Line 14–15 remained relatively high, reaching 32.67 MW, whereas at the scheduling time of 9:15, prior to the fault occurrence on Line 14–15, the flow through Line 14–15 had already decreased to 13.46 MW. Consequently, when the fault occurred on Line 14–15 at 9:28, the power system was not significantly affected by the sudden outage of Line 14–15. Furthermore, at various times from 9:00 to 10:30 before the fault occurred on Line 3–4, the proposed method gradually reduced the flow through this line by appropriately scheduling the generators and energy storage, thereby incrementally increasing the output of internal generation resources in Region B. This ensured that, in the event of a sudden fault on Line 3–4, the internal load in Region B could still receive continuous power supply, thereby averting curtailment resulting from the insufficient ramping and adjustment capability of the internal generation resources, thereby avoiding the phenomenon of power limitation within the region. Consequently, the proposed method adequately considers the impact of line faults on the reduced connectivity of the entire power system, proactively adjusting the output of the generators and energy storage units to reduce the power of the transmission lines connecting the two regions, thereby preventing severe oscillations and widespread transfer of power when a line fault occurs, thereby ensuring the continuity and sustainability of the power supply and enhancing the stability and security of power system operations.

### 4.3. Verification of Virtual-Oscillator-Based Lower-Level Control

In this section, we evaluate the efficacy of the subordinate control approach. In order to align with the aforementioned upper-level preventive control outcomes, we considered the dispatch results of the power system at 9:45 as the initial state for subordinate control and conducted the validation of the proposed voltage- and frequency-based lower-level control method using Region B as the test system. Initially, two frequency events are defined, as described below.

Scenario I: The initial active power of the three energy storage units at Nodes 4, 8, and 14 of Region B are 216.03 MW, 226.24 MW, and 310.31 MW, respectively, which are the dispatch results of the power system at 9:45. The output of the wind farm at Node 9 changes from 400 MW to 300 MW suddenly at *t* = 100 s to emulate the frequency event.

Scenario II: The initial active power of the three energy storage units at Nodes 4, 8, and 14 of Region B are 216.03 MW, 226.24 MW, and 310.31 MW, respectively. The output of the wind farm at Node 9 changes from 300 MW to 335 MW suddenly at *t* = 100 s to emulate the frequency event.

Figure 14 shows the frequency and active power curves of the energy storage units under Scenario I using the proposed lower-level control method. The initial frequency of the microgrid system is 50.00 Hz. Then, the frequency drops dramatically at *t* = 100 s, and the frequency nadirs for the three energy storage units are about 49.885 Hz, 49.865 Hz, and 49.88 Hz, respectively. The frequency starts to recover after the frequency nadir points, and the frequencies of the three energy storage units are basically the same after the transient dynamic processes. The system frequency recovers to 49.92 Hz, 49.96 Hz, and 50.00 Hz at about *t* = 101.8 s, *t* = 104.5 s, and *t* = 120 s, respectively. As for the active power of the energy storage units, the initial values are 216.03 MW, 226.24 MW, and 310.31 MW, respectively. At *t* = 100 s, the active power of the energy storage units increases dramatically, and the stable values of the active power for the three energy storage units are about 236 MW, 248 MW, and 340 MW, respectively. Similarly, Figure 15 shows the frequency and active power curves of the energy storage units under Scenario II using the proposed lower-level control method. In this scenario, the initial frequency of the power system is set at 50 Hz. At *t* = 100 s, the frequency of the three energy storage units suddenly rises to approximately 50.05 Hz. This frequency level, however, is not exceeded, and the frequency then slowly decreases until it returns to the original frequency at around *t* = 120 s. The active power values of the energy storage units decrease significantly at *t* = 100 s, and it takes approximately 1 s to stabilize.

Furthermore, this study examines the performance of energy storage in other initial states of lower-level control. Specifically, one frequency event is defined, which is described below.

Scenario III: The initial active power of the three energy storage units at Nodes 4, 8, and 14 of Region B is 246 MW, 200 MW, and 273 MW, respectively. The output of the wind farm at Node 9 changes from 400 MW to 300 MW suddenly at *t* = 100 s to emulate the frequency event.

Figure 16 shows the frequency and active power curves of the energy storage units under Scenario III using the proposed lower-level control method. At *t* = 100 s, the frequency drops sharply and reaches its lowest point. The frequency begins to recover after reaching the lowest frequency, and the three energy storage units basically return to the original frequency at 120 s. The active power of the energy storage units increases significantly and begins to stabilize after increasing to the maximum value. Scenario III exhibits dynamic processes that are strikingly similar to those in Scenario I and Scenario II, indicating that the proposed approach can effectively regulate energy storage in different initial states. Furthermore, the proposed control method yields satisfactory and stable results in variable control experiments.

### 4.4. Comparison with Droop Control in Lower-Level Control

The output characteristics of the traditional converters of the energy storage units generally simulate the characteristics of a synchronous generator. Depending on the degree of accuracy of the simulation of the synchronous generator, the most representative ones are droop control and VSG control. In order to analyze the difference between the VOC strategy and a traditional control strategy, the droop control strategy has been selected as the compared method in this paper. Droop control is designed to simulate the steady-state droop characteristics of a synchronizer and is a type of proportional control in the phasor domain. Droop control serves as a pertinent comparison to VOC due to its contrasting approach. Characterized by decentralized control, simplicity, and proven performance under varied conditions, droop control serves as a robust benchmark to evaluate the advanced capabilities of VOC. This comparison is instrumental in highlighting the advantages of VOC, such as faster response, enhanced stability, and adaptability to modern grid demands.

In the simulation example, the VOC strategy of the energy storage units in Figure 6 is replaced with the droop control strategy. Meanwhile, considering the differences between the VOC strategy and the droop control strategy, the role of the sparse communication network in the coordination between the VOC strategy and the droop control strategy is temporarily ignored in this paper. That is, when the communication network is removed, all of the energy storage units adopt local control. To compare the VOC strategy and droop control strategy, Scenario IV and Scenario V, described below, were selected to test the two controllers.

Figure 17 and Figure 18 show the active power curves of the energy storage units under Scenario I using both methods. Figure 19 and Figure 20 show the frequency and active power curves of the energy storage units under Scenario II using both methods. By comparing the simulation results, it can be seen that the change process of frequency and active power in the VOC strategy are much faster than those in the droop control strategy. Additionally, the droop control strategy takes a longer time to return to the new stable value, and the maximum overshoot and adjustment time are significantly greater than those of the VOC strategy. Taking Figure 17 and Figure 18 as an example, the VOC strategy is able to stabilize the frequency within 2 s, in contrast to the droop control strategy, where the frequency remains fluctuating even after 20 s of adjustment. Moreover, the VOC strategy effectively avoids overshooting during the adjustment process. As illustrated in Figure 17, under the VOC strategy, the steady-state frequency is approximately 49.85 Hz, with the lowest frequency point being slightly below 49.85 Hz. In contrast, as shown in Figure 18, with the droop control strategy, the system’s frequency fluctuates within a range of about 49.5 Hz to 50.15 Hz, indicating a lower stability in frequency control. This comparison highlights the superior performance of the VOC strategy in maintaining frequency stability and preventing excessive fluctuations, making it a more efficient and reliable control method for power systems.

### 4.5. Influence of the Adaptive Control Parameters in Lower-Level Control

In this paper, an adaptive frequency regulation method with variable parameters is proposed. To clearly capture the indicators measuring the inertial frequency regulation response with different selections of parameters, the frequencies under Scenario IV and Scenario V, described below, using fixed control parameters and adaptive control parameters are compared.

Scenario IV: The initial active power of the three energy storage units at Nodes 4, 8, and 14 of Region B is 216.03 MW, 226.24 MW, and 310.31 MW, respectively. The output of the wind farm at Node 9 changes from 400 MW to 332 MW suddenly at *t* = 100 s to emulate the frequency event.

Scenario V: The initial active power of the three energy storage units at Nodes 4, 8, and 14 of Region B is 216.03 MW, 226.24 MW, and 310.31 MW, respectively. The output of the wind farm at Node 9 changes from 300 MW to 345 MW suddenly at *t* = 100 s to emulate the frequency event.

Figure 21 shows the frequency curves under Scenario IV using the fixed control parameters and the adaptive control parameters. Figure 22 shows the frequency curves under Scenario V using the fixed control parameters and the adaptive control parameters.

It can be observed that both the VOC controllers with fixed and adaptive control parameters enter the transient process at the same time. However, the performance of these two control strategies differs significantly. VOC with adaptive control parameters is able to restore the rated frequency in a shorter period compared to the VOC controller with fixed control parameters. Specifically, the VOC controller with adaptive control parameters takes about 15~20 s to restore the rated frequency, while the VOC controller with fixed control parameters takes much longer. This suggests that the adaptive virtual inertia control strategy is more effective than the traditional VOC strategy in regulating the frequency of the power system. Through the incorporation of virtual inertia elements into the control strategy, VOC controllers with adaptive control parameters are able to respond faster to changes in power fluctuations, resulting in a shorter adjustment time in the frequency response. This advantage is particularly pronounced under transient conditions when the system experiences sudden changes in power demand or supply.

## 5. Conclusions

This paper develops a multi-level operation method for improving the resilience of power systems under extreme weather conditions by incorporating preventive control and virtual oscillator (VO) technology. The method’s ability to precisely estimate the occurrence time of transmission line faults is a significant advancement, enabling the implementation of effective preventive controls. By considering the impact of line faults on system connectivity, the method proactively adjusts the output of generators and energy storage units, effectively reducing the power in transmission lines and preventing severe oscillations during faults. This ensures the continuity and sustainability of the power supply, thereby enhancing the overall stability and security of power system operations. In addition, the method demonstrates robust performance in regulating energy storage under various conditions and shows superior stability and speed in control experiments, particularly highlighting the effectiveness of VOC over traditional droop control. Specifically, compared with droop control, VOC demonstrates the ability to stabilize frequency more quickly, effectively avoids overshooting during the adjustment process, and achieves greater stability in frequency control. These results emphasize the method’s potential as a comprehensive solution for enhancing power system resilience in the face of increasing extreme weather events.

Although the proposed multi-level operation method enhances the resilience of power systems, we must admit that there are challenges in ensuring the stability of adaptive VOC under all possible circumstances. Specifically, our approach may struggle in situations like the tripping of a major generating station. To address this issue, future work could explore new control strategies to improve the system’s adaptability and resilience to sudden events. This could include the development of more advanced predictive models capable of the real-time monitoring and prediction of the behavior of power systems and the automatic adjustment of the VOC strategy upon detecting potential instability signals. For example, by integrating machine learning, it would be possible to design control systems that learn from past failure events and quickly adapt to new scenarios.

## Figures and Tables

**Figure 1 sensors-24-01812-f001:**
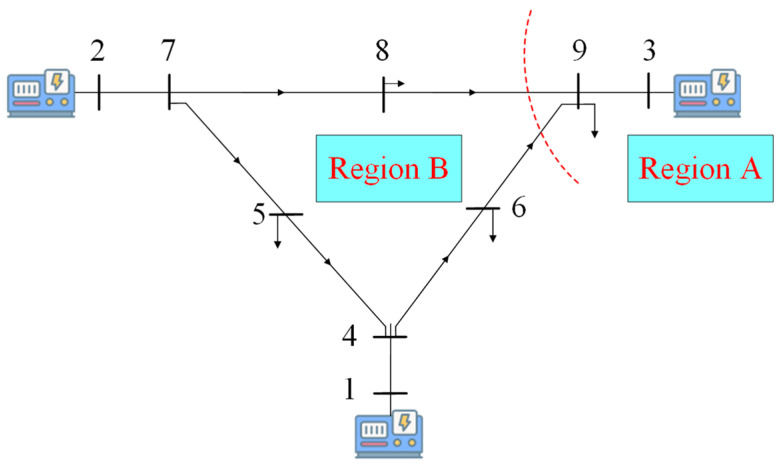
The pre-control process of power flow in a transmission section.

**Figure 2 sensors-24-01812-f002:**
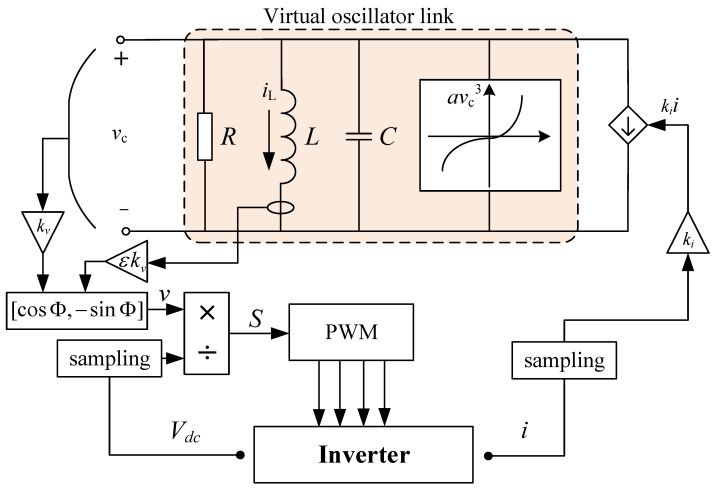
The VdP type of VO.

**Figure 3 sensors-24-01812-f003:**
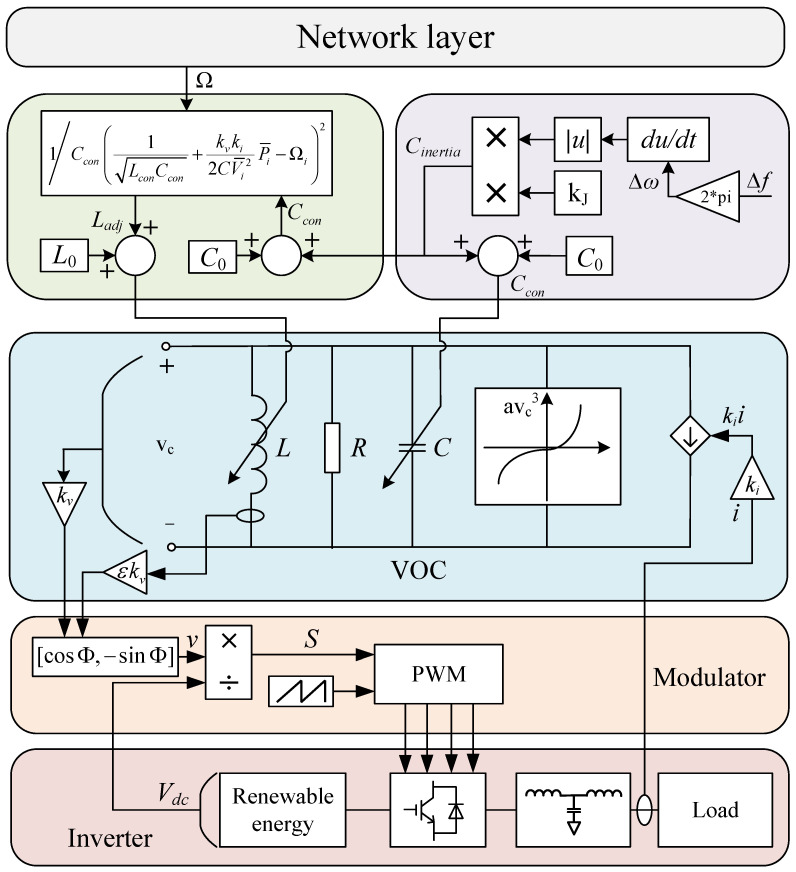
Multi-level control architecture of power system.

**Figure 4 sensors-24-01812-f004:**
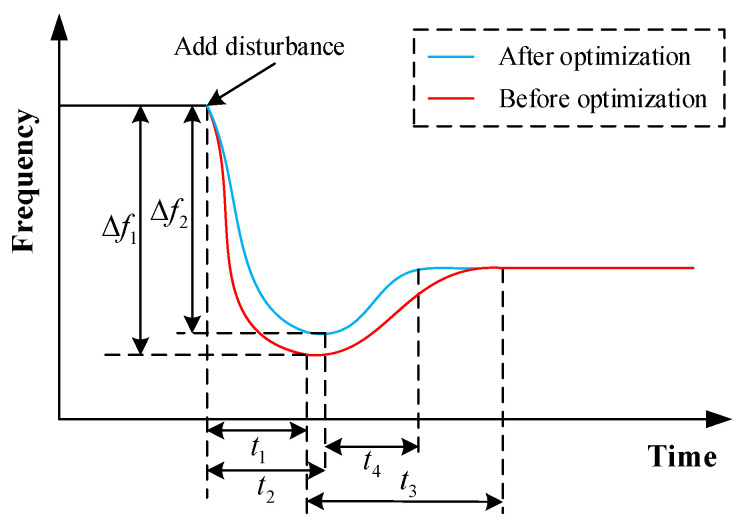
Comparison of transient frequency characteristics of power system before and after ideal inertia optimization.

**Figure 5 sensors-24-01812-f005:**
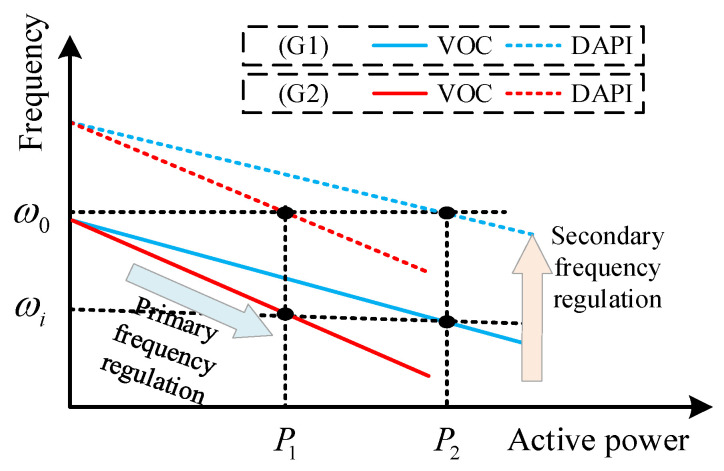
The secondary frequency regulation process based on the VOC droop curve.

**Figure 6 sensors-24-01812-f006:**
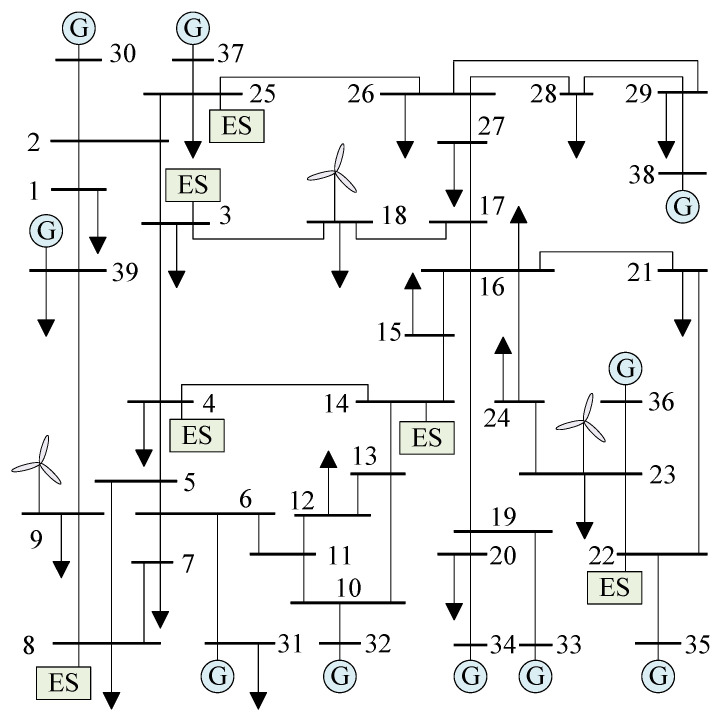
Topology diagram of power system.

**Figure 7 sensors-24-01812-f007:**
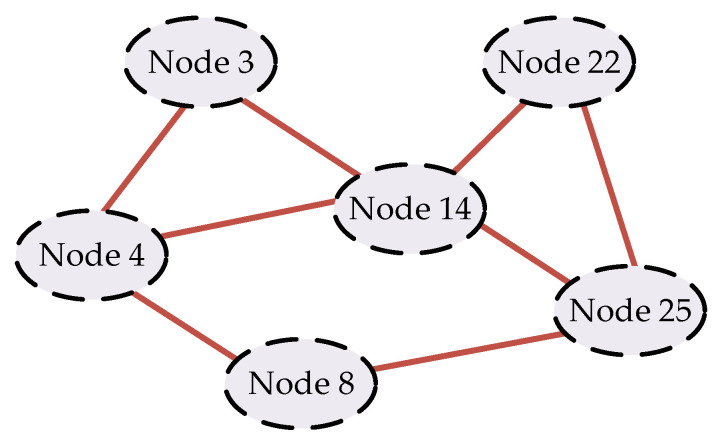
Cyber connection of the energy storage units.

**Figure 9 sensors-24-01812-f009:**
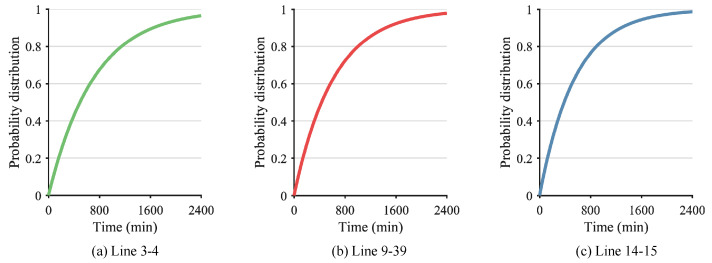
Probability distribution of Line 3–4, Line 9–39, and Line 14–15.

**Figure 10 sensors-24-01812-f010:**
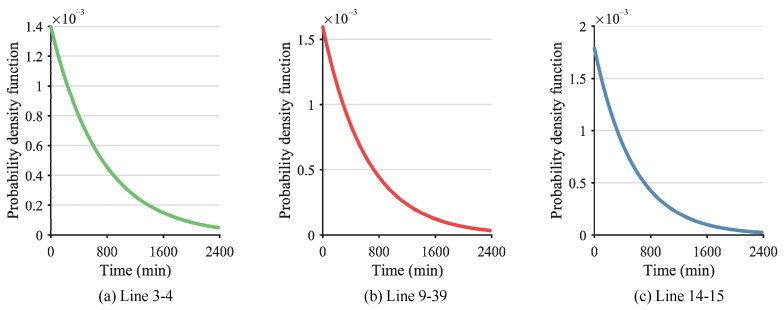
Probability density function of Line 3–4, Line 9–39, and Line 14–15.

**Figure 11 sensors-24-01812-f011:**
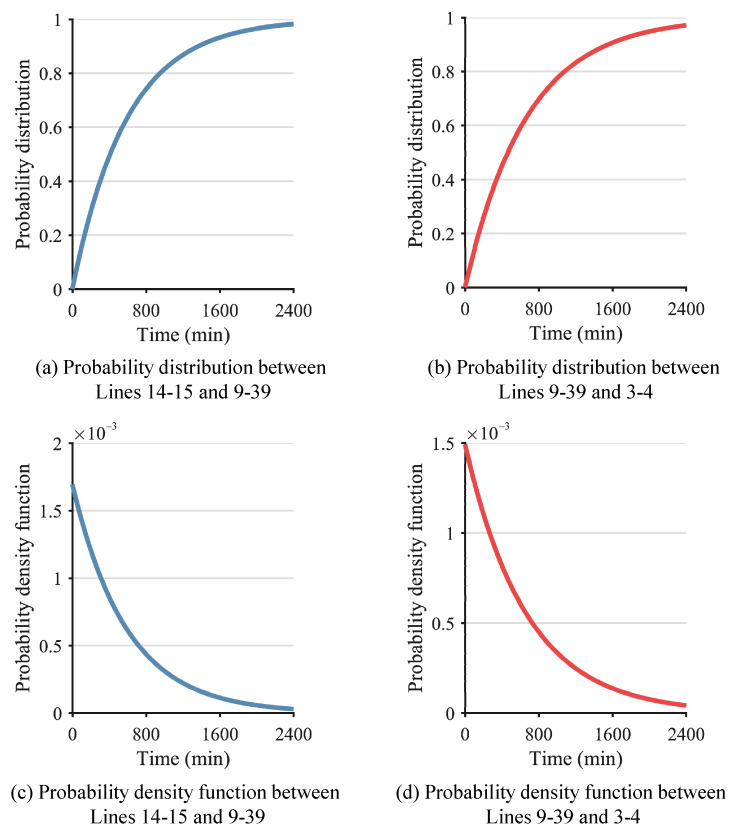
Time interval between successive failures of Lines 3–4 and 9–39 and Lines 9–39 and 14–15.

**Figure 12 sensors-24-01812-f012:**
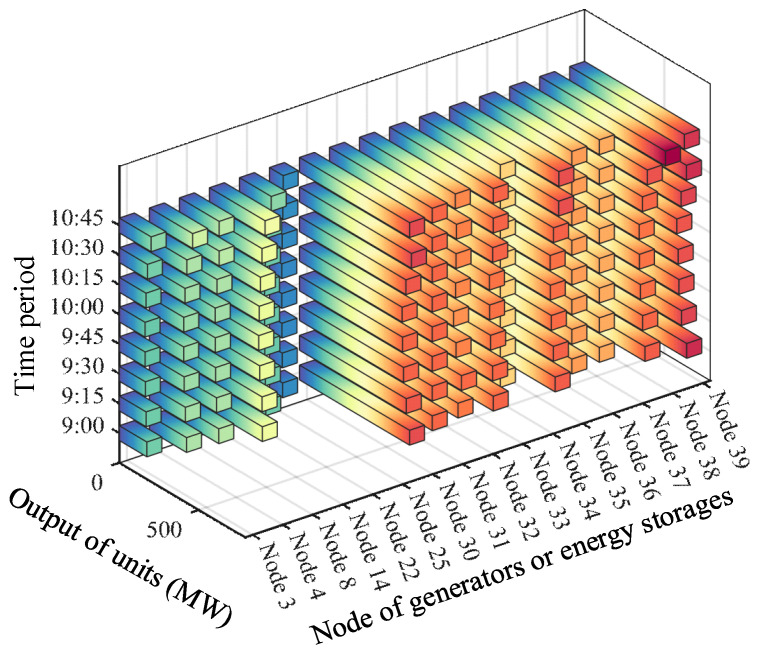
Output of generators and energy storage units in different time periods.

**Figure 13 sensors-24-01812-f013:**
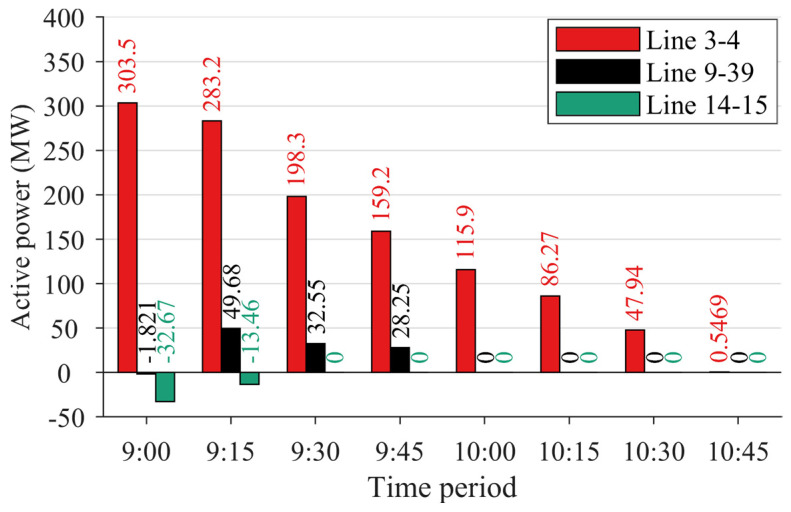
Active power flows in the transmission lines connecting Region A and Region B.

**Figure 14 sensors-24-01812-f014:**
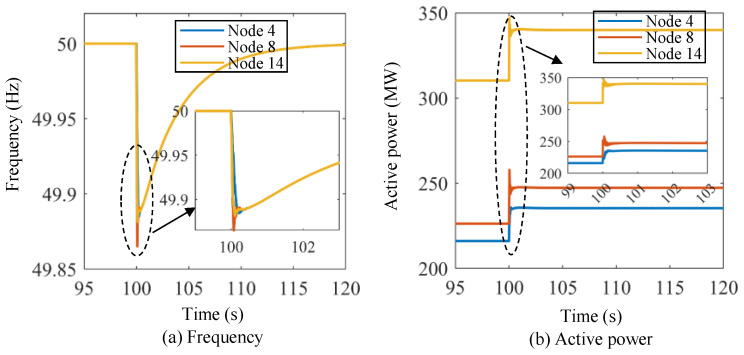
Frequency and active power curves of the energy storage units under Scenario I using the proposed lower-level control method.

**Figure 15 sensors-24-01812-f015:**
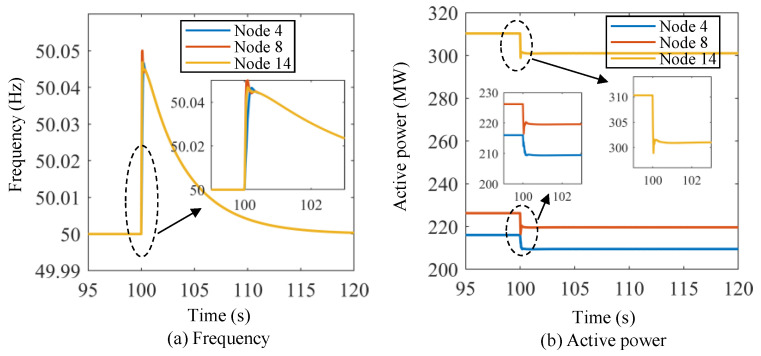
Frequency and active power curves of the energy storage units under Scenario II using the proposed lower-level control method.

**Figure 16 sensors-24-01812-f016:**
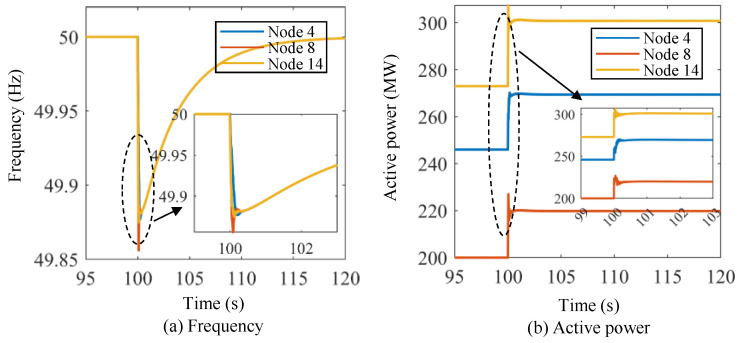
Frequency and active power curves of the energy storage units under Scenario III using the proposed lower-level control method.

**Figure 17 sensors-24-01812-f017:**
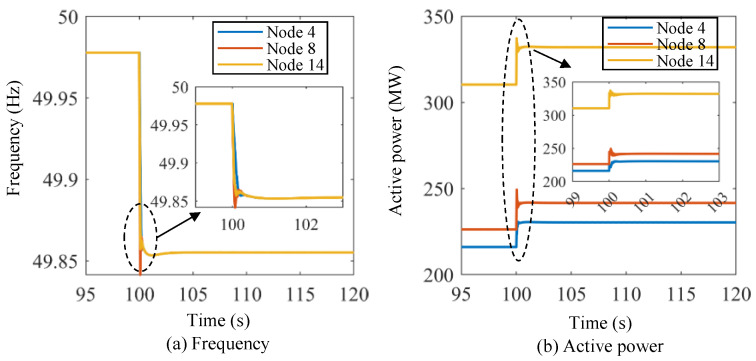
Frequency and active power curves of the energy storage units under Scenario I using the proposed lower-level control method without the communication network.

**Figure 18 sensors-24-01812-f018:**
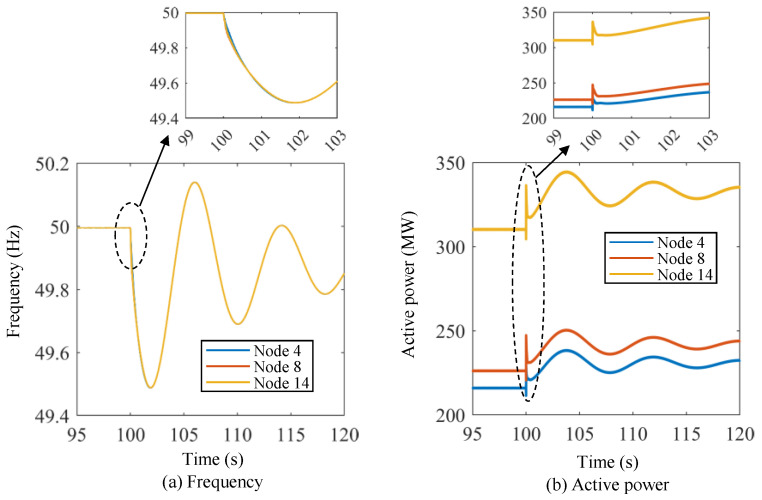
Frequency and active power curves of the energy storage units under Scenario I using the droop-based primary frequency control method.

**Figure 19 sensors-24-01812-f019:**
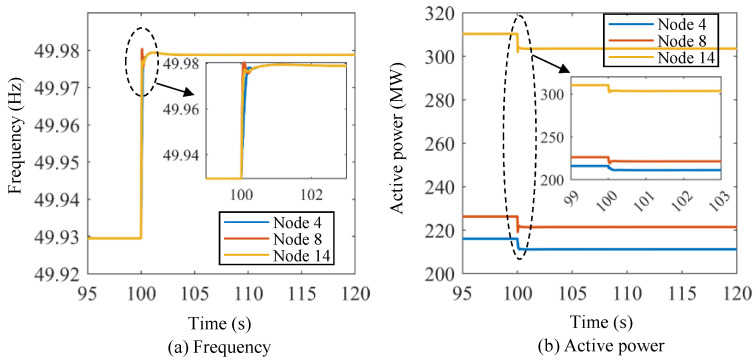
Frequency and active power curves of the energy storage units under Scenario II using the proposed lower-level control method without the communication network.

**Figure 20 sensors-24-01812-f020:**
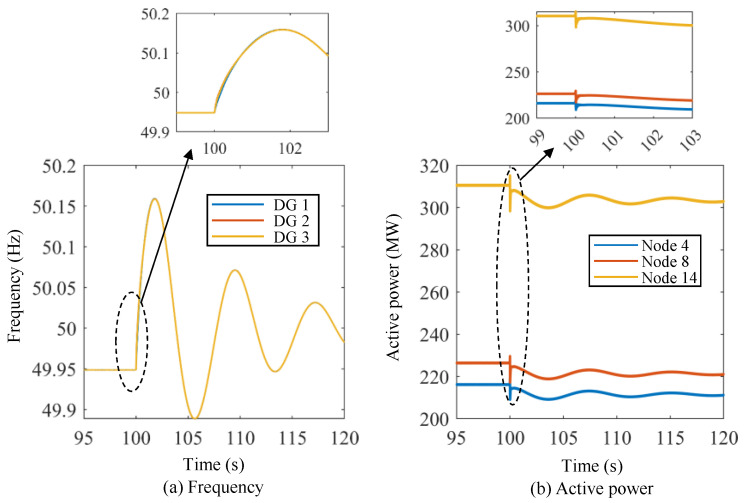
Frequency and active power curves of the energy storage units under Scenario II using the droop-based primary frequency control method.

**Figure 21 sensors-24-01812-f021:**
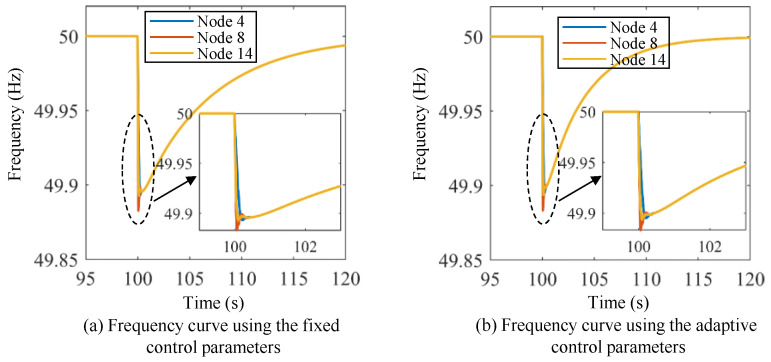
Frequency curves under Scenario IV using the fixed control parameters and the adaptive control parameters.

**Figure 22 sensors-24-01812-f022:**
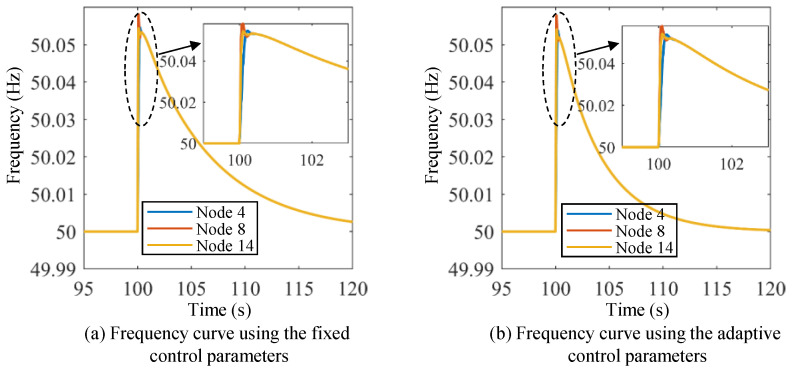
Frequency curves under Scenario V using the fixed control parameters and the adaptive control parameters.

**Table 1 sensors-24-01812-t001:** Parameters of the generators.

Parameter Symbol	Parameter Name and Unit	Value									
		Node 30	Node 31	Node 32	Node 33	Node 34	Node 35	Node 36	Node 37	Node 38	Node 39
$P^{*}$	Upper limit of active power (MW)	1040	646	725	652	508	687	580	564	865	1100
$P^{*}$	Lower limit of active power (MW)	0	0	0	0	0	0	0	0	0	0
$Q^{*}$	Upper limit of reactive power (Mvar)	400	300	300	250	167	300	240	250	300	300
$Q^{*}$	Lower limit of reactive power (Mvar)	140	−100	150	0	0	−100	0	0	−150	−100

**Table 2 sensors-24-01812-t002:** Parameters of the wind farms.

Parameter Symbol	Parameter Name and Unit	Value		
		Node 9	Node 18	Node 23
$P^{*}$	Capacity (MW)	500	400	450
$Q^{*}$	Scale factor	10.7	8.7	10
*L* _f_	Shape factor	3.97	4.7	4.2
*L* _g_	Cut-in wind speed (m/s)	3.5	3.5	3.5
*C* _f_	Rated wind speed (m/s)	15	15	15
$|\Delta \omega {|}_{\max}$	Cut-out wind speed (m/s)	25	25	25

**Table 3 sensors-24-01812-t003:** Parameters of the energy storage units.

Parameter Symbol	Parameter Name and Unit	Value					
		Node 3	Node 4	Node 8	Node 14	Node 22	Node 25
$P^{*}$	Rated active power (MW)	200	300	250	350	200	100
$Q^{*}$	Rated reactive power (Mvar)	20	30	25	35	20	10
*L* _f_	(μH)	300	240	400	300	240	400
*L* _g_	(μH)	300	240	400	300	240	400
*C* _f_	(μF)	20	20	20	20	20	20
*R* _d_	(Ω)	1.5	1.5	1.5	1.5	1.5	1.5
$\omega_{0}$	Rated angular velocity (rad/s)	100π	100π	100π	100π	100π	100π
$|\Delta \omega {|}_{\max}$	Maximum angular velocity offset (rad/s)	π	π	π	π	π	π

**Table 4 sensors-24-01812-t004:** Parameter settings of the VO controllers for the energy storage units.

Parameter Symbol	Parameter Name	Value	Unit
$k_{v}$	Voltage proportionality factor	110	-
σ	Negative impedance	5.0868	Ω^−1^
α	Cubic current source coefficient	3.414	A/V^3^
*L* _0_		58.9	μH
*C* _0_		119.3	mF
$k_{i1}$	The first current scaling factor	0.152	-
$k_{i2}$	The second current scaling factor	0.304	-
$k_{i3}$	The third current scaling coefficient	0.456	-

## Data Availability

Data are contained within the article.

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
