# Peer review of "A Multi-Level Operation Method for Improving the Resilience of Power Systems under Extreme Weather through Preventive Control and a Virtual Oscillator"

_sensors, 2024, doi:10.3390/s24061812_

Round 1

Reviewer 1 Report (Previous Reviewer 3)

Comments and Suggestions for Authors

The quality of the paper has much improved. One of the maim questions was what was the contribution in Modelling successive failures in power systems. This is now moved to appendix and the relevant literature is acknowledged.

Please avoid using superlative expressions like " groundbreaking VOC-based adaptive frequency", as it is up to readers to decide the significance of innovation, authors should point out what is new, and prove the superiority through results, by comparisons with alternatives, etc. 

The answer to question on limitations is not satisfactory, what is provided is some future directions, which are not directly relevant to the research. For example, How would you ensure the stability with adaptive VOC control under all possible circumstances? How would this approach function if some unpredicted event like tripping of a major generating station happens? At least provide some discussion. 

Author Response

Dear Reviewer,

We have carefully considered your insightful comments and suggestions regarding our manuscript. To address your feedback comprehensively, we have prepared a detailed response document. This document outlines how we have aligned our research findings with the evidence and arguments presented, as per your guidance.

Thank you for your valuable input and for guiding us in refining our work. We look forward to your further feedback.

Warm regards,

Ji HAN

Reviewer 2 Report (New Reviewer)

Comments and Suggestions for Authors

The topic addressed in the paper with the title: “A Multi-Level Operation Method for Improving Resilience of Power System Under Extreme Weather Through Preventive Control and Virtual Oscillator” is relevant and very important. It can be applied to more complex systems. However, some improvements should be made.

  1. Section 2 is referred to in line 204, but it is not indicated. Correct it.
  2. Join the four paragraphs into one before section 3.1.
  3. In line 274 check the term negative impedance. Does it not refer to the inverse of resistance?
  4. In Figure 2 improve the title and correct the coefficient of the source current voltage controlled, changing 'a' to 'alpha'.
  5. Correct the signs in equation (12).
  6. In system 12 what is the control input? indicate it.
  7. Clarify the control system synthesis and how it influences the control action. Justify in detailed form equations (14) and (15) form equation (14), where do they come from?
  8. Clarify the sentence in line 346: 'By substituting (16) into equation (15), the following equations can be obtained'. The substitution is not clear.
  9. Complement the title of Figure 6: Topology diagram of IEEE-36 power system.
  10. The active and reactive power symbols in tables 1, 2, and 3 must have suffixes that differentiate them.
  11. In Table 1, the same symbol is used to identify the upper limit and the lower limit of active power. The same thing happens with reactive power. Correct it.
  12. In Table 2, the parameter name column does not correspond to the indicated symbol. Correct it.
  13. In Table 3, the names of the parameters Lf, Lg, Cf, Rd is missing, perhaps they refer to the LCL filter parameters, so it is important to identify each parameter.
  14. In Table 4, identify the names of the parameters L0 and C0.
  15. In line 430 the identification of equations 43 and 48 is not correct.
Comments on the Quality of English Language

  1. In line 17, improve the idea by changing the word demand by supply.
  2. In line 19. describe the idea in singular: the control performance is comprehensively.....
  3. In line 22, the acronym VOC has not been defined.
  4. In line 71 the acronym DG is not defined.
  5. In line 89 improve the phrase:  Advanced frequency regulation methods are required for their stability. It does not have the object.
  6. In the first contribution, at lines 110 and 111 improve the idea, change the word demand by supply.
  7. In line 271 the acronym DG has not been defined.
  8. In line 315 correct the phrase:  the inertia of the frequency regulation dimension.
  9. In line 321 correct the phrase: optimal inertia frequency regulation optimization.

Author Response

Dear Reviewer,

We have carefully considered your insightful comments and suggestions regarding our manuscript. To address your feedback comprehensively, we have prepared a detailed response document. This document outlines how we have aligned our research findings with the evidence and arguments presented, as per your guidance.

Thank you for your valuable input and for guiding us in refining our work. We look forward to your further feedback.

Warm regards,

Ji HAN

Reviewer 3 Report (New Reviewer)

Comments and Suggestions for Authors

The main question addressed by the research is how to enhance the resilience of power systems under extreme weather conditions. The paper specifically focuses on proposing a multi-level operation method that combines preventive control and virtual oscillator (VO) technology to achieve this goal. Additionally, the research aims to develop a novel model for predicting time intervals between successive failures of power systems during extreme weather events. So, consider the topic relevant in the field, and ensure that the conclusions align with the evidence and arguments presented.

Author Response

Dear Reviewer,

We have carefully considered your insightful comments and suggestions regarding our manuscript. To address your feedback comprehensively, we have prepared a detailed response document. This document outlines how we have aligned our research findings with the evidence and arguments presented, as per your guidance.

Thank you for your valuable input and for guiding us in refining our work. We look forward to your further feedback.

Warm regards,

Ji HAN

Reviewer 4 Report (New Reviewer)

Comments and Suggestions for Authors

1.      In Abstract, the abbreviation VOC should be given with the full name.  

2.      System ramping and transmission constraints are mentioned in lines 16 and 86, it is suggested to give a certain explanation.

3.      It is essential to provide an explanation for the variable η in Eq. (9) at line 189.

4.      In the preventive control methods, various constraints are taken into consideration. After discussing all the constraints, it is suggested to provide a concise summary. Additionally, the PSO algorithm is prone to getting trapped in local optima, especially in cases with numerous constraints, if the condition permission, it is suggested to use some new algorithms for solving.

5.      When analyzing the characteristics of the VOC inverter, it is essential to mention the relevant pulse width modulation methods.

6.      In Section 5, Conclusion, it is recommended to detail the effectiveness of VOC compared to the traditional droop control, emphasizing the advantages of VOC.

Author Response

Dear Reviewer,

We have carefully considered your insightful comments and suggestions regarding our manuscript. To address your feedback comprehensively, we have prepared a detailed response document. This document outlines how we have aligned our research findings with the evidence and arguments presented, as per your guidance.

Thank you for your valuable input and for guiding us in refining our work. We look forward to your further feedback.

Warm regards,

Ji HAN

This manuscript is a resubmission of an earlier submission. The following is a list of the peer review reports and author responses from that submission.

Round 1

Reviewer 1 Report

Comments and Suggestions for Authors

The paper is well written, and the whole framework is adequately described. 

There are a few format issues that need to be addressed.

On the whole its a good contribution to the body of knowledge.

Reviewer 2 Report

Comments and Suggestions for Authors

The paper is well presented and topic is the need of time.  

My concerns and suggestions are;

Please include the quantification results in the abstract.

The keywords is not arranged alphabetically.

In line 301, the aim described and is repeated which can be avoided.

For the battery storage, the VOC strategy is replaced with the droop control strategy. Give the  proper justification for this replacement.

The future recommendation is not described in the conclusion section.

Comments on the Quality of English Language

In the paper, minor editing of the English language is required. 

Reviewer 3 Report

Comments and Suggestions for Authors

This manuscript addresses an important subject highly relevant to the industry, but by attempting to cover large scope, manuscript ends-up providing too little details to understand each of the three main concepts, i.e the process of predicting the timing/locations of cascaded tripping, preventive generation dispatch, and control of energy storage systems.

1)     It is not clear how the method of predicting the timing of cascading failures using Poisson model is connected to the physical system and physical processes. The failure probabilities must be related to the change in wind speeds (assuming this is an event triggered by wind related weather event) and the topology and loading on the lines. I would expect the probabilities to dynamically change during the cascading process. The only parameter in the model is “expectation value of Poisson distribution” and it is not clear how these values are selected.

2)     The results provided to validate the accuracy of the proposed model do to not confirm anything. What is the basis of tripping times given in line 418 (page 15)? Not clear how the given probability distribution functions are obtained. This section need major revision.

3)     The optimization process associated with the preventive control (or rather preventive energy management) seems logical. But what type of optimization algorithm was used? How long it needs to run the optimization? How often one need to run the optimization under rapidly changing environment associated with cascading failures?  

4)     The section on the lower-level control based on the VOC techniques is better presented, but it is not clear what is novel in the paper. The performance comparison must be done with a more advance control strategy.  

In general, there needs to be more clarity on what is novel and what is obtained from the previous works. No citations of literature after the introduction. Please provide appropriate references to previous work, equations, models, etc. The is no discussion of the limitations of the proposed methodology.

I would suggest authors to split the paper and focus on one area at a time to better present the paper with solid validations of concepts with good discussions.

Comments on the Quality of English Language

English language is acceptable, minor edits can be made to improve.